# Short-Term Exposure to Benzo(a)Pyrene Causes Disruption of GnRH Network in Zebrafish Embryos

**DOI:** 10.3390/ijms24086913

**Published:** 2023-04-07

**Authors:** Ilaria Gentile, Valeria Vezzoli, Sara Martone, Maria Grazia Totaro, Marco Bonomi, Luca Persani, Federica Marelli

**Affiliations:** 1Department of Medical Biotechnology and Translational Medicine, University of Milan, 20133 Milan, Italy; 2Division of Endocrine and Metabolic Diseases, Laboratory of Endocrine and Metabolic Research, IRCCS Istituto Auxologico Italiano, 20149 Milan, Italyfederica.marelli.fm@gmail.com (F.M.); 3IFOM-FIRC, Institute of Molecular Oncology, 20139 Milan, Italy

**Keywords:** zebrafish, hypolamo-pituitary-gonadal (HPG) axis, gonadotropin releasing hormone (GnRH), endocrine disruptor chemical (EDC), benzo(a)pyrene (BaP)

## Abstract

Benzo(a)pyrene (BaP), a polycyclic aromatic hydrocarbon, is considered a common endocrine disrupting chemical (EDC) with mutagenic and carcinogenic effects. In this work, we evaluated the effects of BaP on the hypothalamo-pituitary-gonadal axis (HPG) of zebrafish embryos. The embryos were treated with 5 and 50 nM BaP from 2.5 to 72 hours post-fertilization (hpf) and obtained data were compared with those from controls. We followed the entire development of gonadotropin releasing hormone (GnRH3) neurons that start to proliferate from the olfactory region at 36 hpf, migrate at 48 hpf and then reach the pre-optic area and the hypothalamus at 72 hpf. Interestingly, we observed a compromised neuronal architecture of the GnRH3 network after the administration of 5 and 50 nM BaP. Given the toxicity of this compound, we evaluated the expression of genes involved in antioxidant activity, oxidative DNA damage and apoptosis and we found an upregulation of these pathways. Consequently, we performed a TUNEL assay and we confirmed an increment of cell death in brain of embryos treated with BaP. In conclusion our data reveal that short-term exposure of zebrafish embryos to BaP affects GnRH3 development likely through a neurotoxic mechanism.

## 1. Introduction

The hypothalamo-pituitary-gonadal (HPG) axis is predominantly regulated by gonadotropin releasing hormone (GnRH) neurons that act with a pulsatile mechanism and stimulate the secretion and synthesis of the two pituitary gonadotrophins, luteinizing hormone (LH) and follicle-stimulating hormone (FSH), which stimulates sex steroids secretion (e.g., estrogen and testosterone) that regulate gonadal development [1]. Several genetic and environmental factors have been described to affect, at many levels, the reproductive functions of vertebrates. Among the factors though to affect reproduction are the exposure to certain chemicals, foods, social environment or stress [2,3].

Zebrafish represent a gold standard method to analyze the costs of environmental injury to reproduction since the endocrine axis that sustain sexual maturation and the surrounding genetic programs are well conserved in teleosts [1]. In teleost species, such as medaka and zebrafish, GnRH3 is detected in terminal nerve, hypothalamus and POA, recapitulating the function of the mammalian GNRH1, probably missed during evolution [4]. It has been shown that environmental toxic compounds, named endocrine disrupting chemicals (EDCs), influence endocrine functions by interfering with synthesis, secretion, transport and metabolism of hormones [5]. In animal models, polycyclic aromatic hydrocarbons (PAHs), well known class of EDCs, have been reported to interfere with reproductive functions by altering hormone balance and production [6]. PAHs are the most ubiquitous pollutants and, among them, Benzo[a]pyrene (BaP) is considered a common EDC with mutagenic and carcinogenic effects [7,8]. Sources EDCs can be very different, since they are introduced into the environment via natural emission (e.g., volcanic eruption) or human activities (e.g., vehicle emission, oil shipping, and refineries); most of them are derived from the incomplete combustion of tobacco and fossil fuels [9]. Despite the efforts to reduce emissions, PAHs concentrations remain above the allowed upper limits in many countries, thus pressing the establishment of their potential toxic effects on human health [10,11]. Different hypotheses have been proposed to elucidate the pathogenic mechanism of PAHs, considering uptake modality, age, duration of exposure, latency of action and dosage. It has been shown that CYP450 activates PAHs into metabolites that are highly reactive with the DNA and, in particular, the ultimate mutagenic product of BaP (BPDE) can generate guanine adducts or can be dehydrogenated by aldo-keto reductase (AKR) to form BaP- catechol that enters in a redox cycle resulting in Reactive Oxygen Species (ROS) production [12,13]. Interestingly, besides oxidative DNA damage, BaP can influence DNA methylation and, depending on its concentrations and the cell type, it causes hypomethylation or hypermethylation of genes involved in carcinogenesis [14].

In mouse models, the exposure to BaP results in reproductive dysfunction, such as reduced production of spermatozoa, lower serum testosterone levels and impairment of oocyte meiotic progression resulting in a low fertilization rate [8,15,16]. In addition, exposure of BaP causes reproductive damages to aquatic organisms, affecting sexual differentiation and hatching time in medaka fish and impacting ovarian development by disrupting oocytes in *Chlamys farreri* [17,18]. In zebrafish, BaP impacts on ovarian development which results in a reduced fertility in female adult fish. Indeed, in adult ovary of female fish, both 17betaestradiol and testosterone levels were reduced, together with impaired hatching and fertilization rate [19]. Interestingly, studies focusing on the impact of BaP on the GnRH3 neurons funtion are lacking, but it was recently described that Bisphenol F, an EDC analogue of bisphenol A, affects GnRH3 neurons in zebrafish by acting through an estrogenic pathway [20]. Gao et al., showed that the embryonic exposure of BaP caused a diminished expression of all of these HPG markers in zebrafish adults [19]. The persisting effects of embryonic exposure of BaP on both juvenile gonadal development and adult reproductive capacity suggest the existence of early deregulation of GnRH3 system. To explore this possibility, in our work we took advantage of the zebrafish transgenic line *tg*(*GnRH3*:EGFP) [1,21] to study the effects of short-term embryonic exposure to BaP on GnRH3 neurons development.

## 2. Results

Zebrafish embryos treated since 2.5 up to 72 hpf with 5 nM or 50 nM of BaP did not show any morphological alterations (Figure 1A–F), nor significant differences in the survival rates when compared to the embryos treated with the control vehicle (1.2% DMSO) (Figure 1G).

We then concentrated on GnRH3 neuron development using the *tg*(GnRH3:EGFP) line [21]. In normal conditions, GnRH3 neurons start to differentiate and proliferate at the level of the olfactory placode (OP) at 24–30 hpf, projecting their axons dorsoventrally towards the pallium and converging at the midline to form the anterior commissure (AC). Around 40–48 hpf, new fibers elongate along the pre-optic area (POA) to innervate the retina (Re) and extend dorsocaudally their axons to reach the hypothalamus (Hy) at 72 hpf. The analysis of GnRH3 architecture was performed in control and BaP-treated embryos at 33, 48 and 72 hpf, spanning the entire GnRH3 neuron development. At 33 hpf, the GFP signal at the level of olphactory bulbs (OBs) and retina (Re) was dramatically decreased in both 5- and 50 nM-treated embryos when compared with the DMSO controls (Figure 2B–D). At 48 hpf, we observed a dose-dependent reduction of GFP on OBs, AC, POA, Re, and Hy innervation in treated embryos (Figure 2F–H): the GnRH3 positive signal was diminished in AC and Re after exposure to 5 nM BaP and almost completely absent after 50 nM BaP treatment (Figure 2M). Nonetheless, at 72 hpf the GnRH3 neuronal architecture appeared completely compromised after the exposure of both 5 and 50 nM BaP (Figure 2J–L).

The dose-dependent reduction of GnRH3-GFP cells were also confirmed by flow cytometry analysis in control and treated embryos at 48 hpf (Figure 3A–D).

Given the mechanism of toxicity of BaP reported in zebrafish and other species [18,22], we evaluated by qPCR the expression of genes involved in antioxidant activity (*sod1*, *sod2*, *cat1*), oxidative DNA damage (*ogg1*) and apoptosis (*casp3a* and *casp3b*), in control and treated embryos at 48 hpf. Embryos exposed to 5 nM BaP exhibited an increased expression of *sod1* and *casp3b* (Figure 4A,F), whereas the expression levels of the antioxidants *sod1*, *sod2* and *cat1* (Figure 4A–C) and of the apoptotic markers *casp3a* and *casp3b* (Figure 4E,F) were significantly upregulated in 50 nM BaP-treated embryos. No changes in the expression of *ogg1* were observed in all conditions (Figure 4D).

At this point, to test whether BaP neurotoxicity was associated with GnRH3 defects we performed a TUNEL assay on embryos at 48 hpf. Compared to the few apoptotic cells detectable at the level of OBs and AC of DMSO-controls (Figure 5B), a strong apoptotic signal was visible in the olphactory and optic regions in 5 nM BaP treated embryos (Figure 5C). Consistently with the GnRH3 phenotype and qPCR results, the embryos treated with 50 nM BaP showed a dramatic increment of cell death in brain, especially in olphactory, preoptic area and hypothalamus (Figure 5D). 

Finally, we evaluated the expression of the main transcripts belonging the HPG-axis in larvae at 120 hpf (Figure 6). Consistently with the reduced fluorescent signals observed in *tg*(*GnRH3*:EGFP) embryos after BaP exposure, the expression of *gnrh3* and of its pituitary receptor *gnrhr3* were significant downregulated after both 5 and 50 nM BaP at larval stage (Figure 6A,B). However, despite the impairment of GnRH3 signal, the levels of *fshb* and *lhb* of BaP-treated larvae appeared comparable to the controls (Figure 6C,D). Regarding the expression of the other transcripts that sustain gonadal development, *vtg1* was upregulated in a BaP dose-dependent manner whereas *esr1* was significantly lower in 50 nM BaP larvae when compared with 1.2% DMSO controls (Figure 6E,G). Lastly, *vtg2* and *esr2* appeared preserved in all groups (Figure 6F,H).

## 3. Discussion

In the present work, we report for the first time the developmental consequences of BaP exposure on GnRH3 architecture using zebrafish embryos as a model system. The analysis of GnRH3 neuronal architecture at different time-points, covering the entire GnRH3 development, revealed a dose-dependent effect of BaP. The severity of GnRH3 phenotypes, ranging from the reduced number GnRH3 fibers at the level of olfactory bulbs and retina to a severe reduction or complete absence of signals in almost all GnRH3 sites. The dose-dependent reduction of GnRH3-GFP cells in BaP-treated embryos was also confirmed by FACS analysis. The lack of gross morphological alterations (including brain necrosis) in treated embryos, confirmed that the GnRH3 phenotypes were not caused by overall embryonic toxicity of BaP.

Several studies described the mechanism of action of BaP based on their uptake, age and exposure window, latency, and dosage [5,22]. The oral exposure to BaP resulted in developmental, immunological, and reproductive effects such as decreased fertility and neurobehavioral alterations in animal models [8,17,19,23,24,25,26,27]. Reproductive defects included the impairment of ovary weight, sperm counts and follicles numbers [8,15,16,17,19,25,27]. Moreover, epidemiological studies in humans reported adverse birth outcomes, including reduced birth and postnatal body weight and lower head circumference, other developmental, neurobehavioral effects that are generally analogous to those observed in animals, providing supportive evidence for hazards associated with BaP exposure [10,23,28,29,30]. However, studies on the interfering effects of BaP on GnRH neuron development are lacking.

To test whether BaP affects GnRH3 development, we investigated the expression of several markers of antioxidant response and apoptosis. Confirming those reported by others [31], BaP caused neurotoxicity as revealed by the increased expression of antioxidant and apoptotic genes. Coherently, embryos treated with 5 or 50 nM BaP exhibited increased cell death in several brain areas, as olphactory placode, retina, and hypothalamus. Finally, accordingly to GnRH3 network impaiment observed in reporter embryos, BaP esposure was associated with a significant dose-depentent downregulation of *gnrh3* and *gnrhr3* transcripts in treated larvae at 120 hpf. Interestingly, this was not accompanied by a downregulation of gonadotrophins, whose expression is directly dependent by GnRH3-GnRH3R signal. These results might be explained by the relative low expression levels of both *fshb* and *lhb*, which start to be expressed at larval stages (Appendix A). It is possible that sensitivity of qPCR is not enough to detect expression changes in these transcripts and that might be evident in later stages. Infact, differences in HPG transcripts levels were previously described in zebrafish adult females exposed to BaP during embryonic window [19]. Brain and ovary samples exhibited, a dose-dependent downregulation of all of these hormones, and as a consequence, a reduced fertility and fecundity rates [19].

The overall data suggest that BaP increased the oxidative stress in GnRH3 neurons that is only partially counteracted by the increment of antioxidants expression. The persistence of stress-environment inevitably caused the activation of apoptotic pathway and neuronal death. Although the carcinogenic effects of BaP received more attention than neurotoxicity, the toxic effects of BaP on nervous system development has been previously described in humans and animal models [23,24,32]. Environmental exposure to BaP correlated with impaired memory and learning ability in humans [23,28]. Studies in rodents also identified the potential point of departure (POD) values for cancer and neurotoxicity endpoints. The POD carcinogenic value in rodents was 0.55 mg BaP/kg-bw-day whereas the POD for neurotoxicity value was markedly lower, 0.025 mg BaP/kg-bw-day [23]. Taking into consideration that neuronal development is a complex process consisting of multiple events that occur during early life it seems reasonable that neurotoxicity is more susceptible to BaP exposure than carcinogenesis. Chepelev and colleagues also hypothesized a preliminary mechanism of action to explain BaP-induced neurotoxicity: (i) BaP binding to the aryl-hydrocarbon receptor (AHR); (ii) AHR-dependent modulation of N-methyl-d-aspartate-glutamate receptor (NMDAR); (iii) NMDAR-mediated neuronal cell death and (iv) compromised learning and memory. However, it is also possible that tissue oxidative damage mediated by BaP may finally lead to either neurotoxicity or carcinogenesis [23,24]. Other studies have also identified epigenetic modification as a BaP-dependent toxic effect. Acute exposure of BaP in Tergillaca granosa is associated with increased antioxidant activity and lipid peroxidation levels that negatively correlates with global methylation [22]. In this context, the oxidative-stress conditions seem to inhibit DNA methylation with the aim of increasing antioxidant capacity against BaP toxicity [22]. Similarly, BaP administration to medaka adults induced changes in germ-cell DNA methylation that caused neurotoxicity and bone deformities in unexposed offspring [23]. In zebrafish, BaP exposure significantly reduced egg production and offspring survival, and induced global hypomethylation [24]. Interestingly, the methylation levels of GnRH3 were elevated in the adult brain, which might be caused by up-regulation of DNA methyltransferase 1 and 3a [17]. Since DNA methylation is established from 4.3 to 6 hpf and that organogenesis is completed at 96 hpf, it reasonable that DNA methylation patterns would be easily changed by BaP [24].

In our study, GnRH3 development was affected after embryonic exposure to BaP for a short period, likely caused by neurotoxicity. However, defects in DNA methylation profiles should be considered as an additional or intermediate factor for BaP toxicity.

## 4. Materials and Methods

### 4.1. Zebrafish Line and Maintenance 

All experiments were performed according to EU regulations on laboratory animals (Directive 2010/63/EU). The zebrafish studies were approved by the Body for the protection of Animals (OPBA) of the University of Milan, Italy (protocol 198283).

Transgenic line (*GnRH3*:EGFP) was obtained by the Gothilf Lab (Tel Aviv University, Tel Aviv, Israel) and embryos were maintained in a flow-through system at a constant temperature (28 °C), with a photoperiod (light:dark) of 14:10. Zebrafish embryos were obtained from natural spawning and raised until the desired developmental stages according to established morphological criteria. Starting from 24 h post fertilization (hpf), embryos were harvested in fish water containing 0.002% of 1-phenyl-2-thiourea (PTU; Sigma-Aldrich, St. Louis, MO, USA) to prevent pigmentation and 0.01% methylene blue to prevent fungal growth.

### 4.2. BaP Treatment

The working doses of BaP (5 and 50 nM) were used and administered as previously reported by Gao et al. and our group [19,33]. For each experiment, pools of 100 embryos were raised in glass petri dish (diam. 150 mm; Merck KGaA, Darmstadt, Germany) and treated with 100 mL of BaP solution at a concentration of 5 nM and 50 nM in fish water (Sigma-Aldrich, St. Louis, MO, USA) prepared freshly from stock solutions of 0.1 mg/mL and 1 mg/mL respectively. Treatments were performed from 2.5 to 72 hpf and the BaP solutions were renewed twice a day. As control, embryos were treated with 1.2% of dimethyl- sulfoxide (DMSO, Sigma-Aldrich, St. Louis, MO, USA), used as solvent for BaP solution.

### 4.3. RNA Extraction and Quantitative PCR (qPCR)

Total RNA was extracted from pools of 30–40 embryos treated with BaP or DMSO at 48 hpf using TRIzol Reagent (Thermo Fisher Scientific, Waltham, MA, USA). cDNA synthesis reaction was carried out following the protocol of GoScript Reverse Transcription System (Promega, Madison, WI, USA). Quantitative real-time PCR (qPCR) was performed by ABI PRISM 7900HT Fast Real-Time PCR System using SYBR Green Master Mix (Thermo Fisher Scientific, Waltham, MA, USA). Beta-actin gene was used as endogenous control. Each experiment were performed in triplicate, using pools of embryos derived from three independent treatment. Results are presented as mean ± SEM. Statistics were analysed using ONE way ANOVA with Tukey post-hoc correction. *** *p* < 0.001; ** *p* < 0.01; * *p* < 0.05; ns = not significant. The primers used in this study are listed in the Appendix A.

### 4.4. Immunofluorescence (IF)

*Tg*(*GnRH3*:EGFP) embryos at desided developmental stages were fixed in 4% paraformaldehyde (Sigma Aldrich, St. Louis, MO, USA) ON at 4 °C. Fixed embryos were permeabilized with cold Acetone for 15 min, washed five times with PBT (PBS with 0.1% Tween20) and rinsed for 30 min in NH4Cl. Embryos were then incubated for 3 h in blocking solution (5% BSA in PBT), and ON a 4 °C with mouse anti-GFP (1:500, Merk-Millipore, Burlington, MA, USA). After several wash in PBT, embryos were raised for 2 h in blocking solution followed by ON incubation with goat anti-mouse IgG Alexa Fluor 488-conjugated (1:1000, Thermo Fisher Scientific, Waltham, MA, USA). IF-embryos were mounted in glass slides and acquired by confocal microscope (A1 HD25/A1R HD25 instrument, Nikon FRET-FLIM) provided by the UniTech nolimits NOxsz < LIMITS service (University of Milan, Milan, Italy).

### 4.5. Flow Cytometry Experiments

Flow cytometry experiments were performed on *Tg*(*GnRH3*:EGFP) zebrafish embryos at 48 hpf. Embryos dissociation and intracellular straining for GFP was performed as described [34,35]. Mouse anti-GFP (1:500, Merk-Millipore, Burlington, MA, USA) and goat anti-mouse IgG Alexa Fluor 555 conjugated (1:1000, Thermo Fisher Scientific, Waltham, MA, USA) were used as primary and secondary antibodies, respectively. Acquisitions were performed using Attune NxT instrument equipped with 4 lasers (405, 488, 561, 633 nm) (Thermo Fisher Scientific, Waltham, MA, USA). Analysis were done using Kaluza Software (Beckman Coulter, Brea, CA, USA). AB wild-type embryos were used to set the gate to exclude cells auto-fluorescence.

### 4.6. TUNEL Assay

Apoptotic cells in *tg*(*GnRH3*:EGFP) were evaluated by using the Invitrogen Click-iT Plus TUNEL Assay for In Situ Apoptosis Detection, Alexa Fluor 647 dye (Thermo Fisher Scientific, Waltham, MA, USA) following manufacturer’s instructions. Embryos were fixed at 48 hpf in 2% PFA and dehydrated in absolute methanol overnight at −20 °C. Embryos were then rehydrated, washed with PBS and permeabilized for 35 min with proteinase-K. Embryos were then fixed in 2% PFA 20 min, washed in PBT and treated 10 min with TdT reaction buffer. After incuba-tion with TdT reaction cocktail for 1 h at 37 °C embryos were washed in 3% BSA in PBT and incubated in Click-IT reaction cocktail 30 min at RT. Embryos were finally washed six times in PBT over 4 h and mounted in glycerol for confocal acquisition.

## Figures and Tables

**Figure 1 ijms-24-06913-f001:**
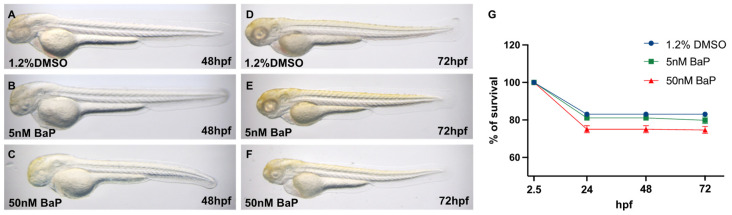
Embryonic morphology at 48 or 72 hpf treated with (**A**–**D**) 1.2% DMSO; (**B**–**E**) 5 nM BaP; (**C**–**F**) 50 nM BaP. BaP administration did not cause significant alterations in the body plan of zebrafish embryos. (**G**) Survival rates of 1.2% DMSO controls and 5 and 50 nM BaP treated embryos at different developmental stages. Results are reported as % derived from three independent experiments with at least 100 zygotes each.

**Figure 2 ijms-24-06913-f002:**
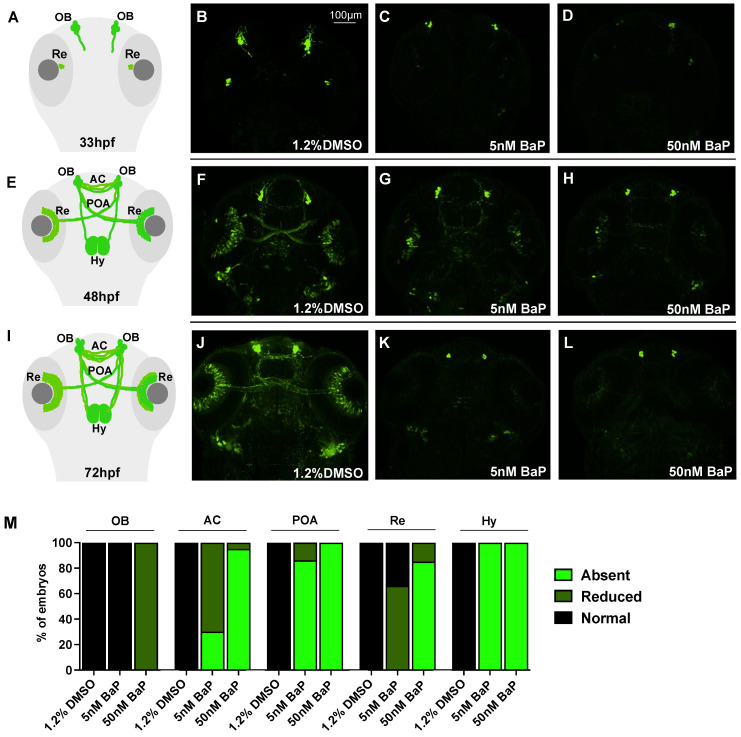
Schematic representation of GnRH3 neurons development at 33 (**A**) 48 (**E**) and 72 hpf (**I**). Confocal images of *tg*(*GnRH3*:EGFP) at 33 (**B**–**D**) 48 (**F**–**H**) and 72 (**J**–**L**) hpf. Embryos treated with BaP revealed a dose-dependent reduction of GFP signal at the level of OP, AC, Hy, POA, and Re compared to 1.2% DMSO control embryos. OP: olfactory placode; AC: anterior commissure; POA: pre-optic area; Re: retina Hy: hypothalamic innervation. Embryos were acquired mounting the heads in dorsal view and each experiment was performed in triplicate using about 30 embryos/stage. (**M**) Histogram showing the frequency of embryos that express normal, reduced or absent GFP fluorescence in 1.2% DMSO and BaP treated embryos at 48 hpf.

**Figure 3 ijms-24-06913-f003:**
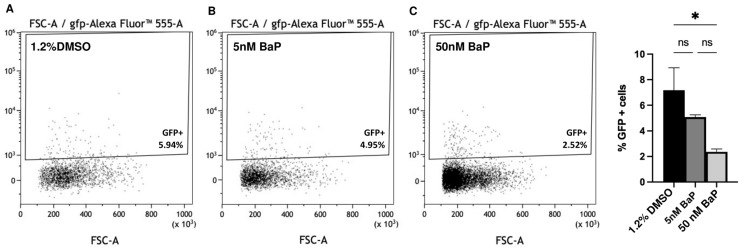
Quantification of the GFP+ cells by flow cytometry analysis of the GnRH3 neuron cells (**A**–**C**). The % of GFP+ cells are reported in (**D**). ONE way ANOVA with Tukey post hoc correction. Results are presented as mean ± SEM. * *p* < 0.05; ns = not significant. Experiments were performed in triplicates using pools of 1.2% DMSO, and BaP treated embryos at 48 hpf.

**Figure 4 ijms-24-06913-f004:**
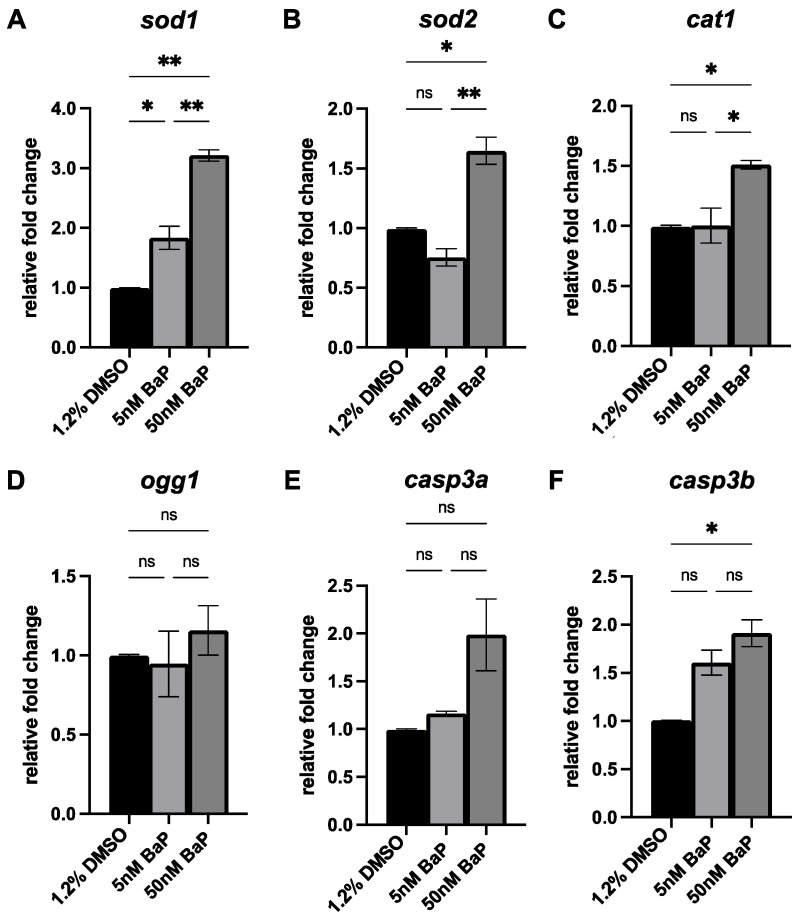
qPCR of *sod1* (**A**), *sod2* (**B**), *cat1* (**C**), *ogg1* (**D**), *casp3a* (**E**) and *casp3b* (**F**) in embryos treated with 5 and 50 nM BaP and their relative controls at 48 hpf. Results are presented as mean ± SEM. ** *p* < 0.01; * *p* < 0.05; ns = not significant.

**Figure 5 ijms-24-06913-f005:**
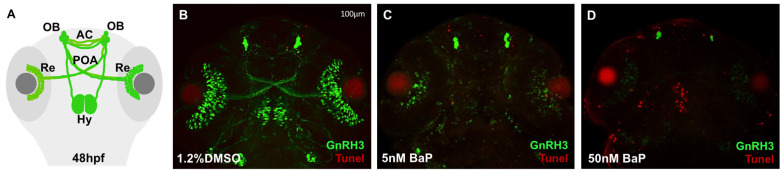
Representation of GnRH3 neurons development at 48 hpf (**A**). Confocal images of TUNEL assay on *tg*(*GnRH3*:EGFP) at 48 hpf in embryos treated with 1.2% DMSO, 5 and 50 nM BaP (**B**–**D**). Embryos were acquired mounting the heads in dorsal view and each experiment was performed in triplicate using about 30 embryos/stage.

**Figure 6 ijms-24-06913-f006:**
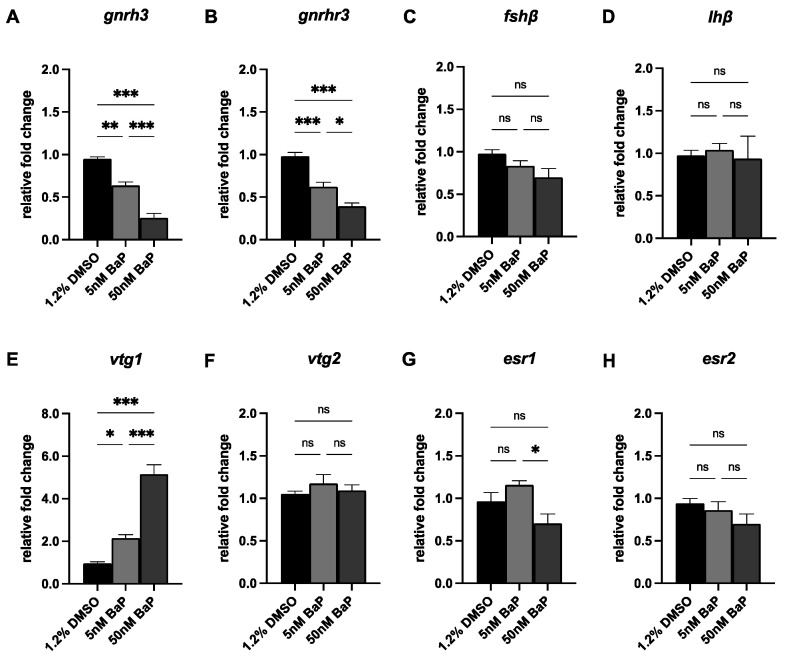
qPCR of *gnrh3* (**A**), *gnrhr3* (**B**), *fhsb* (**C**), *lhb* (**D**), *vtg1* (**E**), *vtg2* (**F**), *esr1* (**G**) and *esr2* (**H**) in 120 hpf larvae treated with 5 and 50 nM BaP and 1.2% DMSO controls. Results are presented as mean ± SEM. *** *p* < 0.001; ** *p* < 0.01; * *p* < 0.05; ns = not significant.

## Data Availability

Raw data of the study are available at the following link: https://zenodo.org/record/7808139#.ZC_p3C8QPq0 (accessed on 22 February 2023).

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
