# Peer review of "Short-Term Exposure to Benzo(a)Pyrene Causes Disruption of GnRH Network in Zebrafish Embryos"

_ijms, 2023, doi:10.3390/ijms24086913_

Round 1
Reviewer 1 Report
1. Table 1: Gene accession number should be provided for the genes studied. Beta actin sequence is not included in the table.
2. Confocal images should be counter stained with DAPI. How did the authors prepare the sample for immunofluorescence? The procedure is not explained.
3. Flow cytometry data is not convincing.
4. Counterstain for the TUNEL assay is not there. Fig. 5D, the eye of zebrafish fluorescence seems like more when compared to others – may be the authors adjusted the brightness of the picture.
5. What is the significance of using a transgenic line over a normal fish? GnRH3 antibody could be used to detect the GnRH3 protein in normal zebrafish.
6. Because Benzo(a)Pyrene treatment decreased GnRH3 mRNA levels, it cannot be concluded as it specifically affects the GnRH3 and likely acts as a neurotoxin. Besides, why did not the expression levels of other genes affect with the treatment?
7. Did the author study any withdrawal effects?
Author Response
- Table 1: Gene accession number should be provided for the genes studied. Beta-actin sequence is not included in the table.
We thank the reviewer for the comment. We correct the Table 1 accordingly (Now Supplementary Table 1 in the revised version).
- Confocal images should be counter stained with DAPI. How did the authors prepare the sample for immunofluorescence? The procedure is not explained.
We included the full IF procedure in the revised version methods (Line 351-360). Regarding the counterstaining we performed the staining with DAPI (see attached file). Whole embryos were incubated for 20 minutes at RT in 1mM DAPI dissolved in embryo medium (CaCl2•2H2O · 0.49 g, 0.33 mM; MgSO4•7H2O · 0.81 g, 0.33 mM; Methylene blue, 10 g, 0.1% (w/v)). In the images included below, you can see that DAPI stains all the head structures and providing non-informative results because it interferes with the lower fluorescence GFP or TUNEL signals. For that reason, we have not included these figures in the revised version of the manuscript.
- Flow cytometry data is not convincing.
We redid the histogram presenting the FACS data as mean ± SEM. Statistical test that we have used (ONE way ANOVA with Tukey post hoc correction) confirmed that the GFP+ cells in 50nM BaP embryos were significantly lower than 1.2% DMSO (p<0.05), which pointed to our hypothesis that BaP affects GnRH3 development in a dose-dependent manner.
- Counterstain for the TUNEL assay is not there. Fig. 5D, the eye of zebrafish fluorescence seems like more when compared to others – may be the authors adjusted the brightness of the picture.
All embryos were acquired with the same confocal parameters and the images were not corrected for brightness or contrast. However, it is true there is some variability in fluorescence even between embryos belonging the same group. We redid the Figure 5 selecting DMSO controls, 5 nM and 50 nM BaP embryos (derived from the same experiment) with comparable fluorescence in the lens. We did not include the counterstaining for the reasons explained in Q2.
- What is the significance of using a transgenic line over a normal fish? GnRH3 antibody could be used to detect the GnRH3 protein in normal zebrafish.
Tg(GnRH3:GFP) line allows the visualization of GnRH3 development in real-time and with high resolution, as reported by various research groups over the years (Abraham et al., 2009; Wang et al,. 2011; Bassi et al., 2020; Mancini et al., 2020; Zuccarini et al., 2020; Cotellessa et al., 2023). Instead, GnRH3 antibody was reported in just one paper, which is used for GnRH3 analysis in brain sections of adult fish (Spicer et al., 2016).
- Because Benzo(a)Pyrene treatment decreased GnRH3 mRNA levels, it cannot be concluded as it specifically affects the GnRH3 and likely acts as a neurotoxin. Besides, why did not the expression levels of other genes affect with the treatment?
GnRH3 development is completed by 3 days post-fertilization (dpf) whereas the expression of the downstream genes belonging the HPG-axis appears later in development. For instance, FSHb and LHb expression, which are directly regulated by GnRH3, starts to be expressed in larval stages (Zhang et al, 2012; Cheng et al., 2012; Nica et al, 2006) and for that reason the RNA levels at 120 hpf of are very low. This is evident by the Ct values of fshb and lhb that are around 32-33 and by the electrophoresis of qPCR fragments (Supplementary Figure 1).
It is reasonable to think that sensitivity of qPCR is not enough to detect differences in the expression of these transcripts, which might be evident in later stages. Unfortunately, due to Italian Laws, in this study we cannot performed zebrafish experiments after 5 dpf but we planned to analyze the effects of BaP on juveniles and adult sexual development and reproductive functions in the future.
In the revised version, we include the analysis of gnrh3 by qPCR (Figure 6) and the electrophoresis of qPCR fragments (Supplementary Figure 1), and a comment in the discussion (lines 268-276).
- Did the author study any withdrawal effects?
We thank the reviewer for the suggestion. We did not study any withdrawal effects of BaP in this work and no other authors have reported there are any after BaP treatment. However, the analysis withdrawal effects take time and several additional experiments, but this could be a possible future perspective of our next work.

Reviewer 2 Report
The manuscript is a high-quality scientific work that is devoted to the study of the mechanisms of the influence of Benzo(a)pyrene on the hypothalamic-pituitary-gonadal axis of zebrafish embryos. The authors' data show that short-term exposure to Benzo(a)pyrene in zebrafish embryos affects the development of GnRH3 through a neurotoxic mechanism.
There are a few notes about this work:
1. In figure 2, the figure M has a very different font from other figures. In addition, it is very small and hard to read. In my opinion figure M should be increased.
2. In Figure 3, the text in figure captions and on the axes is very small. It should be unified. Make it the same as in the other pictures. This is especially true for the values along the axes - they are not readable at all.
Author Response
We thank the reviewer for the comments. We redid figure 2 and 3 accordingly.

Round 2
Reviewer 1 Report
The authors have improved the manuscript significantly and answered well for the reviewer's comments.